# Induced Pluripotent Stem Cell-Derived Cardiomyocytes Therapy for Ischemic Heart Disease in Animal Model: A Meta-Analysis

**DOI:** 10.3390/ijms25020987

**Published:** 2024-01-12

**Authors:** Quan Duy Vo, Yukihiro Saito, Kazufumi Nakamura, Toshihiro Iida, Shinsuke Yuasa

**Affiliations:** 1Department of Cardiovascular Medicine, Faculty of Medicine, Dentistry and Pharmaceutical Sciences, Okayama University, Okayama 700-8558, Japan; dr.duyquan@gmail.com (Q.D.V.); pqwy461x@s.okayama-u.ac.jp (T.I.); yuasa@okayama-u.ac.jp (S.Y.); 2Department of Cardiovascular Medicine, Okayama University Hospital, Okayama 700-8558, Japan

**Keywords:** induced pluripotent stem cell, ischemic heart disease, outcomes, safety, meta-analysis

## Abstract

Ischemic heart disease (IHD) poses a significant challenge in cardiovascular health, with current treatments showing limited success. Induced pluripotent derived–cardiomyocyte (iPSC-CM) therapy within regenerative medicine offers potential for IHD patients, although its clinical impacts remain uncertain. This study utilizes meta-analysis to assess iPSC-CM outcomes in terms of efficacy and safety in IHD animal model studies. A meta-analysis encompassing PUBMED, ScienceDirect, Web of Science, and the Cochrane Library databases, from inception until October 2023, investigated iPSC therapy effects on cardiac function and safety outcomes. Among 51 eligible studies involving 1012 animals, despite substantial heterogeneity, the iPSC-CM transplantation improved left ventricular ejection fraction (LVEF) by 8.23% (95% CI, 7.15 to 9.32%; *p* < 0.001) compared to control groups. Additionally, cell-based treatment reduced the left ventricle fibrosis area and showed a tendency to reduce left ventricular end-systolic volume (LVESV) and end-diastolic volume (LVEDV). No significant differences emerged in mortality and arrhythmia risk between iPSC-CM treatment and control groups. In conclusion, this meta-analysis indicates iPSC-CM therapy’s promise as a safe and beneficial intervention for enhancing heart function in IHD. However, due to observed heterogeneity, the efficacy of this treatment must be further explored through large randomized controlled trials based on rigorous research design.

## 1. Introduction

Approximately one-third of global deaths are attributed to cardiovascular diseases [1]. Among these, ischemic heart disease (IHD) stands out as the primary contributor to cardiovascular morbidity and mortality in both developed and developing countries [2]. According to the 2019 Global Burden of Disease report, ischemic heart disease was responsible for 49.2% of all deaths related to cardiovascular diseases [3].

Ischemic heart disease is characterized by the substantial loss of cardiomyocytes following a sudden reduction in myocardial perfusion. Nonetheless, no endogenous repair mechanisms have proven sufficient for restoring the lost myocardial tissue or reviving cardiac function. Consequently, the loss of myocardial tissue triggers a cascade of events, leading to the development of a non-contractile scar, thinning of the ventricular wall, and heart remodeling, ultimately resulting in heart failure and death [4]. However, the effectiveness of the available management for IHD remains quite limited. Despite intensive medical treatment with statins, beta-blockers, angiotensin-converting enzyme inhibitors (ACEi) or angiotensin receptor blockers (ARB), and antiplatelet agents, the all-cause mortality rates of IHD patients were 57.5 per 1000 person-years [5]. The sole available treatment option for replacing the loss of cardiac muscle is cardiac transplantation. However, this approach is constrained by the limited supply of donors and the necessity of lifelong immunosuppressive therapy. It is worth noting the requirement that cardiomyocyte transplantation, as elaborated in the following section, currently involves the use of immunosuppressive drugs. The aspiration is that future technological progress will tackle this hindrance.

Recently, regenerative medicine involving stem cell treatment to restore damaged heart tissue has emerged as a promising management approach for ischemic heart disease (IHD). In the early 21st century, Takahashi et al. successfully transformed somatic cells into pluripotent stem cells using retrovirus and introduced Oct3/4, Sox2, c-Myc, and Klf4 [6]. Unlike ESCs, iPSCs have the advantage of being free from ethical issues and rejection in the case of autologous transplantation. Furthermore, their functionality remains unaffected by the aging process [7]. The capacity to generate cardiomyocyte tissue, triggering the heart’s contractile movement, presents an advanced approach in the field of regenerative medicine. Despite the potential utility of undifferentiated induced pluripotent stem (iPS) cells in therapy, substantial evidence indicates the risk of tumor formation in both control and IHD rats, regardless of the administered dose [8]. Various methods have been developed to remove undifferentiated cells that form tumors from differentiated cells, and the results of animal studies have demonstrated that the risk of tumor formation is quite low.

Many studies have indicated that cardiomyocytes derived from induced pluripotent stem cells (iPSC) demonstrate the formation of spontaneously beating sarcomeres during in vitro culture [9,10]. The evaluation of action potential characteristics has shown that iPSC-derived cardiomyocytes (iPSC-CM) exhibit sensitivity to β-adrenergic stimulation and possess differentiation potential for ventricular, atrial, and nodal cardiomyocyte lineages [11]. In vivo, the administration of iPSC-CM has effectively integrated with the host myocardium, significantly reducing fibrosis, and substantially increasing fractional shortening in a murine model of ischemic heart disease (IHD) [12]. Moreover, iPS cell-derived bioengineered tissue has recently shown the ability to enhance cardiac contractility compared to controls following myocardial infarction (MI) [13]. Numerous preclinical studies involving animals have individually explored various conditions, such as the timing of transplantation, animal species, and methods of inducing ischemia. However, there has been a lack of comprehensive analysis considering the collective outcomes of these experiments. To address this gap, we conducted a meta-analysis to assess the current evidence regarding the efficacy and safety outcomes of iPSC-CM treatment in IHD models. The result of this review aims to provide recommendations and evidence-based insights that can guide future human trials.

## 2. Results

### 2.1. Search Results

In the initial search, a total of 1808 references were identified. Subsequently, 1602 references were excluded during the first screening process. The remaining 196 potential references underwent abstract review, followed by 70 articles in full-text review. After careful consideration, 51 articles were included. A PRISMA flow diagram is provided in Figure 1.

#### 2.1.1. Characteristics of Included Studies

A comprehensive meta-analysis incorporated a total of 51 articles. The publication dates of these articles spanned from 2011 to 2023. Among the 51 studies, 43 were conducted on small animals (murine), while 8 involved large animals (porcine, primates, rabbit), comprising a total of 1012 animals (516 in treatment groups and 496 in control groups). Notably, no human trials were identified. Further information on the studies is outlined in Appendix A.

#### 2.1.2. Intervention Characteristics

In 51 trials, the cryo-injure model was employed in three studies, the infarct–reperfusion model in five studies, and a permanent ischemic heart disease model was utilized in forty-four studies. Most studies adopted hiPSC-derived cardiomyocytes (N = 44, 86.27%). The predominant method of cell delivery involved intramyocardial cell injections (n = 35, 66.03%), while bio-engineered tissue treatment was implemented in 18 studies (33.96%), and intra-coronary injection was utilized in 1 study (0.01%). The total doses of injected cells varied from 2 × 10^5^ to 4 × 10^8^. Follow-up duration ranged from 1 to 12 weeks.

#### 2.1.3. Risk of Bias Assessment of Included Studies

None of the studies met all ten criteria for a low risk of bias. Only one research study satisfied seven criteria for a low risk of bias, while another fulfilled six criteria. Across all studies, low bias risk was identified concerning comparable baseline groups, complete data, and selective reporting (Figure 2 and Appendix A).

### 2.2. Safety of iPSC-CM Treatment

#### 2.2.1. Mortality

Mortality outcomes were either documented or could be accurately determined in a total of 19 studies (Figure 2) [9,10,12,14,15,16,17,18,19,20,21,22,23,24,25,26,27,28,29,30]. There was no difference in the mortality risk between the iPSC-CM and control groups, with OR = 0.61; 95% CI: 0.3 to 1.24, *p* = 0.17 (Figure 3).

#### 2.2.2. Arrhythmia

Four studies examined the potential risk of arrhythmia following iPSC-CM treatment, comprising one study involving small animals [9] and three studies involving large animals [19,30,31]. No significant difference was observed in the occurrence of sustained ventricular arrhythmias between the cell-based group and the control group, with an odds ratio of 10.46, a 95% confidence interval ranging from 0.94 to 115.83, and a *p*-value of 0.06 (Figure 4).

### 2.3. Efficacy of iPSC-CM Treatment

#### 2.3.1. Ejection Fraction (EF)

The meta-analysis revealed a significantly higher left ventricular ejection fraction (LVEF) during follow-up between iPSC-CM therapy and the control group, with a mean difference (MD) of 8.23 (95% CI, 7.15 to 9.32; *p* < 0.001). There was notable heterogeneity (*p* < 0.001) and high inconsistency (I^2^: 95%) in the small animal studies (Figure 5).

#### 2.3.2. Fractional Shortening (FS)

In 25 studies assessing fractional shortening (FS), which involved 428 small animals and 17 large animals, iPSC-CM treatment demonstrated improvement compared to controls, with a mean difference (MD) of 5.16 (95% CI 4.25 to 6.08; *p* < 0.001), accompanied by high heterogeneity (*p* < 0.001) and high inconsistency (I^2^: 95%) (Figure 6).

#### 2.3.3. Other Cardiac Outcomes

The implementation of iPSC-CM showed a tendency to decrease left ventricle fibrosis in the iPSC-CM treatment group compared to the control group. The cell-based treatment group exhibited a trend toward reducing left ventricular end-systolic volume (LVESV) and end-diastolic volume (LVEDV), although the difference was not statistically significant. It is essential to highlight the presence of significant heterogeneity (*p* < 0.001) and substantial inconsistency (I^2^ > 90%) observed across the studies (Table 1).

Figure 7 illustrates the summary of outcomes.

#### 2.3.4. Subgroup Analyses

In our subgroup analysis, we investigated the impact of various factors such as animal size, delivery method, treatment timing, disease model, and follow-up duration on the effectiveness of iPSC-CM treatment in inducing cardiac changes (Appendix A). The analysis indicated a trend toward more significant enhancements in response to cell therapy, particularly during the 4–8 week follow-up period. Beyond this time point, the impact of cell therapy appears to decrease. Additionally, late injection of cells after myocardial infarction (>1 month) and using bioengineered tissue resulted in better improvements. In contrast, the ischemia/reperfusion myocardial infarction model demonstrated less benefit compared to chronic infarction models (Table 2).

#### 2.3.5. Meta-Regression

A sensitivity analysis was initially conducted to investigate the causes of heterogeneity in the outcomes of ejection fraction. This analysis identified three outliers characterized by effect estimates that were more than 1.5 times the interquartile range (IQR) from the median [15,32,33]. These outliers were notable for their exceptionally high effect estimates relative to the rest of the data. Upon their exclusion, the recalculated overall effect estimate and heterogeneity were assessed for the modified dataset. This recalibration resulted in a slight decrease in the Weighted Mean Effect Estimate, from 8.12 to 8.05. However, a significant level of heterogeneity persisted, with an I^2^ value of 94.26%, only marginally reduced from the previous 94.45%. Subsequently, a meta-regression analysis was conducted. The findings of this analysis, as detailed in Table 3, indicated that certain factors, specifically the method of delivery (intracoronary and intramyocardial injection) and the origin of the cells (xenogeneic), had a significant impact on the effect estimates (the coefficient was −8.065, 3.824, and 2.014). Additional details and results about meta-regression analysis on FS, LVESV, LVEDV, and LV fibrosis area are available in the Appendix A (Appendix A).

## 3. Discussion

Ischemic heart disease, a major contributor to global morbidity and mortality, arises from an imbalance between myocardial oxygen supply and demand [58]. In advanced stages of IHD, the effectiveness of revascularization and medical therapy may diminish [59]. Stem cell therapy has emerged as a novel approach to enhance cardiac function in patients with advanced ischemic heart failure [60]. This form of treatment has the potential to improve tissue perfusion, promote the growth of new blood vessels, and preserve or regenerate myocardial tissue [61]. Our review conducted a meta-analysis of cell-based therapies used in treating animal models with IHD. The primary outcomes encompassed safety and efficacy. Safety measures were categorized into mortality and the occurrence of adverse events. Efficacy was assessed through various indicators of cardiac function, with a focus on the preservation of ejection fraction. To obtain deeper understanding of the factors influencing the effect size on ejection fraction, subgroup analysis was conducted, taking into account variables such as the delivery route, timing of administration, disease model, and follow-up time.

Our research consistently demonstrates the effectiveness of iPSC-CM treatment in enhancing LVEF by 8.33% (7.18 to 9.47) in small animals and 8.38% (4.03 to 12.72) in large animal studies, with no significant heterogeneity observed between these groups. These results are in concordance with findings from several prior studies. The 2011 meta-analysis by Van der Spoel T.I.G et al., focusing on large animal models, assessed the impact of stem cell therapy on ischemic cardiomyopathies and reported a significant LVEF improvement of 7.51% (95% CI: 6.15% to 8.87%) [55]. Furthermore, a 2022 meta-analysis by Debora La Mantia et al. also reported an LVEF improvement of 7.41% (95% CI: 6.23 to 8.59%) [62]. Additionally, the meta-analysis by Peter Paul Zwetsloot encompassing both small and large animal models showed an overall effect of cardiac stem cell treatment on LVEF of 10.7% (95% CI: 9.4 to 12.1) compared to the control group [63]. However, when interpreting research findings from animal models, it is essential to recognize the considerable differences between small and large animal models. For example, small animals like rodents have high heart rates, with a mouse’s heart capable of beating up to 800 times per minute. This contrasts sharply with larger animals, such as rabbits and porcine species, which have much slower heart rates of 130–300 beats per minute and 50–116 beats per minute, respectively [64]. To maintain cardiac output at these high rates, smaller species require quicker cardiac contractions and relaxations than their larger counterparts. Moreover, the cardiac kinetics of small animals differ from humans due to variations in excitation, calcium handling, myofilament protein isoforms, and genetic characteristics [65]. In contrast, the myocardium of larger animals more closely approximates that of human hearts, often making them more suitable for studies related to human cardiac function [66]. Understanding these differences is important for the accurate interpretation of research data and ensuring the relevance of these findings to human conditions, as they can significantly influence experimental results.

Our findings also revealed that the most significant improvement in LV function occurred within 4 to 8 weeks post-treatment; however, the effectiveness of the iPSC-CM treatment diminished beyond this period. This trend of attenuation is also evident in clinical trials. The BOOST trial, an open randomized study that involved intracoronary injection of mononuclear cells (MNCs) in 30 patients with ST-elevation myocardial infarction (STEMI), showed LVEF and enhanced systolic function after a six-month follow-up. However, at the 18-month follow-up, the improvement in LVEF was no longer statistically significant [67,68]. This outcome suggests that while stem cell therapy may initially enhance the recovery of LVEF after ischemic events, sustained, long-term treatment is necessary to preserve these therapeutic benefits.

The delivery timing has emerged as a critical determinant affecting ejection fraction in ischemic heart disease (IHD). Nevertheless, over the past two decades, establishing the optimal timeframe for cell transplantation has been a contentious issue due to challenges in enhancing cell recruitment and survival. While the transplantation time-window period varies in pre-clinical and clinical studies, the majority lean toward a period within one-week post-acute myocardial infarction (AMI). Our review revealed that cell therapies administered immediately after the disease model induction exhibited a preservation of ejection fraction by 8.21%. Similar outcomes were observed in other pre-clinical studies. For instance, Hu et al. reported a time-dependent therapeutic effect, with the highest preservation in ejection fraction observed in rats receiving intramyocardial mesenchymal stem cell (MSC) injection one week after disease induction [69]. In an IHD rat model, James D. Richardson et al. demonstrated that the group receiving intracoronary cardiac stem cells within 7 days post-induction exhibited a better ejection fraction than the group receiving treatment after 1 week [70]. These findings were further supported by a meta-analysis of 34 clinical RCTs conducted by Xu et al., indicating that administering autologous bone marrow stem cells 3 to 7 days post-percutaneous coronary intervention significantly improved left ventricular function in acute myocardial infarction patients [71]. Remarkably, even though our review encompassed only two studies that utilized iPSC-CM treatment following one month of IHD induction in animals with heart failure (EF < 50%) [21,34]. The analysis also revealed that cell therapies administered one month after myocardial infarction induction resulted in the preservation of ejection fraction by 15.28%. This result aligns with those of YiHuan Chen et al., where implanting mesenchymal stem cells from bone marrow between 2 to 4 weeks following a myocardial infarction proved to be more advantageous in reducing the scar area, preventing left ventricular remodeling, and enhancing the restoration of heart function in a porcine model [72]. The meta-analysis of Debora La Mantia et al. in the large animal model also indicated that the outcomes of cell therapy were more favorable within the time-period 31–60 days after ischemic induction [73]. These findings indicate a potential application of iPSC-CM in the late phase of myocardial infarction, specifically in heart failure patients.

We also investigated the influence of various routes of stem cell administration on preserving left ventricular function, including intracoronary, intramyocardial, intravenous, and via bio-engineered structures. The analysis revealed that bio-engineered tissue treatment exhibited the most favorable effect on ejection fraction in pre-clinical studies, demonstrating a mean difference of 9% (7.2 to 10.80). Supporting these findings, a study by Yu Jiang et al. indicated that the transplantation of bio-engineered tissue resulted in a superior increase in left ventricular ejection fraction and fractional shortening compared to intramyocardial transplantation of hiPSC-CMs in a rat model after myocardial infarction [29]. In addition, our review also highlighted the significant impact of intramyocardial treatment on heart function, with a mean difference of 8.09% (6.79 to 9.39). This aligns with a meta-analysis of clinical studies on bone marrow cell therapy, suggesting that the intramyocardial delivery of stem cells enhances treatment effectiveness in individuals with ischemic heart disease [74,75]. Our analysis identified a negative effect, a decrease of 5.05% (ranging from −10.74 to 0.65), after the intracoronary administration of iPSC-CM. This suggests a potential risk for exacerbating cardiac conditions with this treatment method. In contrast, intravenous therapy seemed to positively impact left ventricular function, showing a 2.34% improvement (ranging from −0.74 to 5.43). However, it is important to note that the observed differences in these treatment approaches did not attain statistical significance. Interpretation of these results should be approached with caution, since they are derived from a single study evaluating the efficacy of these delivery methods [30]. A major challenge in stem cell therapy is ensuring the effective delivery of cells to the targeted injury site. The variable success rates in pre-clinical trials may be attributed to insufficient engraftment at the intended site. Thus, developing more precise cell delivery techniques is a critical objective in the field of cardiac regenerative medicine. Our findings suggest a preference for the application of bio-engineered structures to improve cardiac function. Traditional systemic delivery methods, like intravenous injection, are simple but lack precision in directing cells to specific target areas. which often lead to a significant number of cells relocating to other organs, notably the lungs [76]. Similarly, the efficacy of intracoronary delivery methods is also constrained, as evidenced by only approximately 5% of cells remaining at the transplantation site between 24 to 48 h post-procedure and less than 1% surviving beyond 4 to 6 weeks [77]. However, it is important to note that intramyocardial delivery involves open-heart surgery, a more invasive procedure compared to standard cardiac catheterization. These insights emphasize the necessity for ongoing clinical trials to determine the most effective methods for stem cell delivery. From a clinical standpoint, in cases of acute myocardial infarction, cell-based therapies can be administered via intravenous or intracoronary routes after coronary revascularization, offering a less invasive option. However, for patients with chronic myocardial ischemia who are not candidates for coronary revascularization, direct intramyocardial injection may be preferable. This approach enables the precise targeting of the injection site, potentially enhancing the treatment’s effectiveness.

Our results indicated a less pronounced improvement in LVEF in the ischemia/reperfusion model compared to the chronic occlusion models. These results align with the findings in meta-analyses in both small [63] and large animal models [78]. Moreover, xenogeneic cell therapy (hiPS-CM) showed a better effect on left ventricular function than allogeneic cell therapy. The challenges associated with the functional decline of autologous stem cells due to aging and constrained immediate availability underscore the need for alternative approaches. The allogeneic and xenogeneic cell products is a promising alternative to address these challenges. This outcome instills optimism regarding the potential use of xenogeneic cell replacement as a therapeutic intervention for a wide range of human diseases.

Finally, we assessed the safety of administering iPSC-CM therapies in pre-clinical models of ischemic heart disease. Overall, the administration of iPSC-CM was found to be generally safe, with no significant differences observed in animal mortality or the occurrence of arrhythmia across the studies included in the review. This result aligns with other meta-analyses conducted by Mary Thompson et al. and Manoj M. Lalu et al. on mesenchymal stem cell (MSCs) treatment, which showed no heightened risk of death, malignancy, or adverse events compared to control groups [79,80].

Our research showed significant heterogeneity among the studies. This heterogeneity is particularly important in the context of conducting meta analysis in animal research. We hypothesized that the observed high heterogeneity may stem from a variety of factors including differences in animal models (small versus large), measurement methodologies, timing of injections, follow-up periods, and a diverse range of administration routes and windows. Additionally, heterogeneity could arise from variances in biological study characteristics, such as species, sex, and age. We acknowledge that this diversity could potentially lead to misinterpretations if not adequately addressed. Despite our efforts in implementing sensitivity analysis, multivariable regression analyses, and subgroup analysis to mitigate heterogeneity, we observed no substantial change in the outcome heterogeneity. We contemplated excluding certain studies from the effect size calculation, but recognized that unwarranted exclusions might exacerbate the bias inherent in the meta-analysis. It is worth noting that the heterogeneous nature of animal models has been documented in other research about stem cells in animal models [62,81]. These findings highlight the necessity for more standardized reporting protocols in animal studies to ensure consistency and reliability.

In summary, iPSC-CM therapy appears to be a promising, potentially safe, and effective treatment option for patients with IHD. Presently, the application of iPSC-CM has been restricted to animal models, which poses challenges in directly assessing its efficacy in human patients. To address this gap, several clinical trials are underway to evaluate the effectiveness of iPSC-CM therapy in a clinical context (Table 4). The outcomes of these studies have the potential to pave the way for a new era in the treatment of one of the most debilitating diseases globally.

## 4. Materials and Methods

### 4.1. Meta-Analyses

This research adhered to the Preferred Reporting Items for Systematic Reviews and Meta-Analyses (PRISMA) guidelines, and the final report was compiled with the PRISMA checklist (http://www.prisma-statement.org/, accessed on 1 November 2023.).

#### 4.1.1. Search Strategy

We conducted a literature search using databases, including PUBMED, ScienceDirect, Web of Science, and the Cochrane Library from inception until October 2023, using the following terms and synonyms for “regenerative therapy”, “induced pluripotent stem cell”, and “ischemic heart disease”. After eliminating duplicate records, papers were screened based on their titles and abstracts, and eligibility was assessed through a full-text review conducted independently by two investigators. A third investigator was consulted should there be any discrepancies. Additionally, individual searches within the reference lists of the included studies were conducted to identify any additional research for potential inclusion. No restrictions were imposed on publication dates or languages.

#### 4.1.2. Inclusion and Exclusion Criteria

The inclusion criteria were limited to studies examining the safety and effectiveness of iPSC-CM therapy in ischemic heart diseases: randomized controlled trials (RCTs), clinical and pre-clinical studies with ischemic heart disease models. No constraints were imposed on the route of administration or the source of iPSCs.

Exclusion criteria included studies that used cell types other than iPSC-derived myocardiocytes and studies that used engineered iPSCs to modify the expression of specific genes (except for imaging purposes). Studies not classified as randomized controlled trials (RCTs), secondary reports, poster presentations, reviews, editorials, or studies not presented in English were also excluded.

### 4.2. Outcome Definition

The primary focus of our study was on the efficacy of iPSC therapy, measured by assessing changes in left ventricular ejection fraction (EF), fractional shortening (FS), left ventricular end-systolic volume (LVESV), left ventricular end-diastolic volume (LVEDV), and infarct size.

Our secondary objective was to evaluate the safety of iPSC therapy, assessed by examining the frequency of mortality, arrhythmia, teratoma, or the adverse events related (AE) to the administration of iPSC-CM.

### 4.3. Data Extraction

Two investigators independently gathered and cross-checked data for accuracy, with a third investigator consulted to resolve any discrepancies. Data extraction included bibliographic information (authors’ names, year of publication, funding, title, language, and journal), study design (disease model, objectives, sample size, inclusion/exclusion criteria), animal characteristics (species, age, gender, and immune status), and cell therapy details (cell type, tissue source, cell dose, delivery method, timing, and frequency). Additionally, characteristics of iPSCs, such as passage number, isolation method, and any positive surface markers mentioned, were collected.

As defined earlier, all measurements related to primary and secondary outcomes were recorded. Data were sourced from text, graphs, and plots, with the WebPlot digitizer tool version 4.6 used when specific values were not explicitly stated in the text. In cases of missing or unclear data regarding primary or secondary outcome measures, the corresponding author of the respective study was contacted for information. Furthermore, if an investigation did not explicitly use a particular variable but provided measurements allowing its calculation, the variable of interest was computed using the appropriate equation.

Efficacy outcomes were assessed through the improvement of cardiac function and reduction in infarct size. Safety outcomes were appraised using measures such as all-cause mortality, arrhythmia, and teratoma formation. Cardiac outcome data, including parameters like EF, FS, LVESV, and LVEDV, were extracted at each follow-up point to track changes over time and assess the duration of iPSC-CM treatment.

### 4.4. Quality Assessment

All studies meeting the inclusion criteria were independently evaluated by two investigators for potential bias. Criteria from the Cochrane Handbook for Systematic Reviews of Interventions were applied to randomized controlled trials [82]; while the SYRCLE bias assessment tool was used for animal studies [83]. Assessments were categorized as “low risk of bias”, “high risk of bias”, or “unclear risk of bias”. Overall quality was then categorized as either “low risk of bias” or “high risk of bias”, with an overall “low risk of bias” indicating that all assessment domains were rated as anything but “high risk of bias”.

### 4.5. Statistical Analyses

A meta-analysis was conducted using a random effects model to estimate the effectiveness of iPSC therapy. Risk ratios (RR) with 95% confidence intervals (CI) were calculated for dichotomous outcomes using the Mantel–Haenszel method. For continuous data, the weighted mean difference (WMD) was used if outcomes were measured similarly, and the standardized mean difference (SMD) was used when studies reported the same outcomes but in different ways. In cases where studies had more than two treatment groups, the analysis focused on the iPSC and control groups. Subgroup analyses were conducted to examine the variability in the safety and effectiveness of iPSC therapy based on factors such as route, dose, timing, source, and IHD model. Baseline functional measures were also compared with measurements taken at all time points after iPSC administration.

Statistical heterogeneity between studies was assessed using I^2^, with a value exceeding 50% indicating significant heterogeneity. Subgroup analysis was performed to identify potential factors contributing to heterogeneity. All statistical tests were two-sided, and significance was considered if the *p*-value was less than 0.05. Data were presented as mean values with a 95% confidence interval (95%CI) in forest plots and meta-regression analyses. Statistical analyses were conducted using Cochrane’s software, Review Manager (RevMan) version 5.4.

## 5. Conclusions

The application of iPSC-CM therapy demonstrates a consistent safety profile in pre-clinical animal models of IHD. This treatment also plays a crucial role in preserving heart function. However, the effectiveness of this treatment is influenced by numerous factors such as the delivery route and delivery timing, dosage, and the source of the administered cells.

The findings of this review have significant implications, offering methodological recommendations for ongoing and future clinical studies. Understanding the impact of variables such as delivery methods, timing, dosage, and cell origin can guide researchers in optimizing the design and conducting clinical trials involving iPSC-CM therapies for ischemic heart diseases, ultimately advancing the development and application of iPSC-CM therapies in cardiac treatment.

### Limitations

Our study has certain limitations. The identified parameters such as optimal delivery method, cell dosage, treatment duration, and cell origin in our research might not represent the ideal therapeutic conditions. Instead, they reflect the practices commonly adopted by researchers in the field. To accurately determine the most efficacious therapeutic strategies, it is imperative for future studies to extensively investigate and refine these variables, thereby uncovering their genuine therapeutic potential.

Another limitation is the substantial heterogeneity observed among the studies. This variation arises from numerous factors such as the diversity in cell number, cell type, and animal species used across the studies, leading to variations in the data collected from different comparison groups. This diversity could result in uncertainty about the long-term effectiveness of iPSC-CM treatment in ischemic heart disease. To address this challenge, future research should prioritize conducting structured randomized controlled trials (RCTs), aiming to enhance consistency and reproducibility by the adoption of standardized protocols that control internal validity. This approach would encompass standardizing elements such as cell source, dose, and treatment timing, as well as conducting multicenter animal studies to ensure more reliable and broadly applicable results.

## Figures and Tables

**Figure 1 ijms-25-00987-f001:**
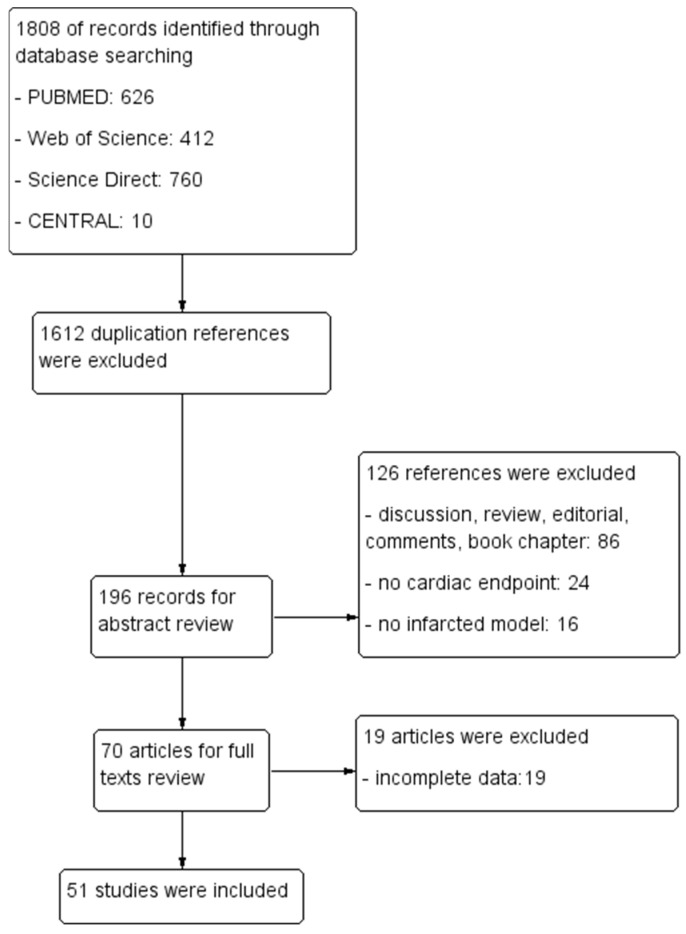
Flow chat demonstrating the current study’s selection process.

**Figure 2 ijms-25-00987-f002:**
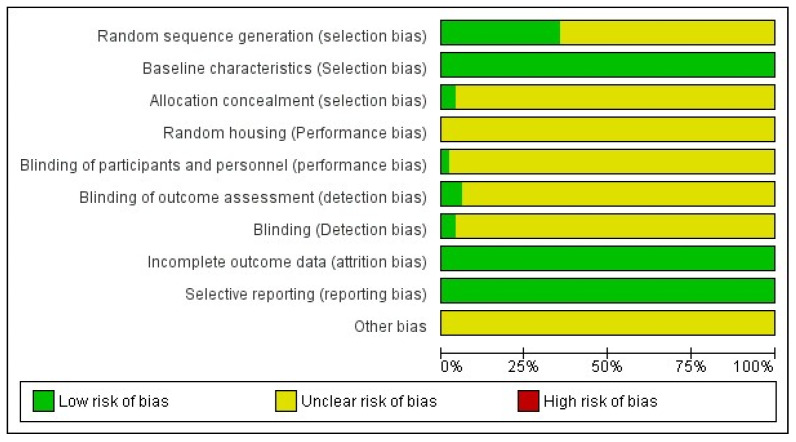
Quality assessment of included articles.

**Figure 3 ijms-25-00987-f003:**
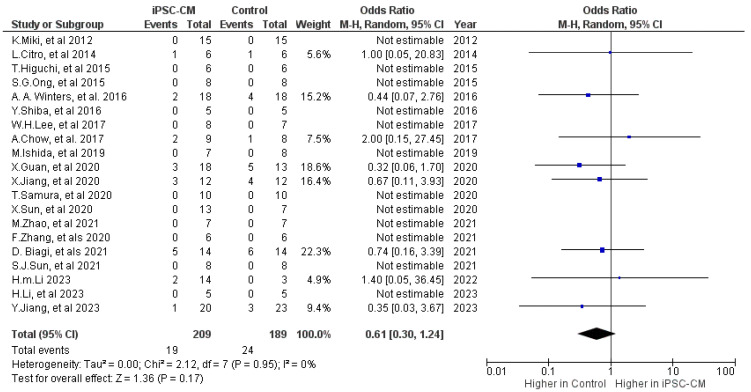
Effect size of iPSC-CM treatment on mortality [9,10,12,14,15,16,17,18,19,20,21,22,23,24,25,26,27,28,29,30]. Blue box represents the point estimate of the effect for a single study, the black diamond represents the overall effect estimate of the meta-analysis.

**Figure 4 ijms-25-00987-f004:**
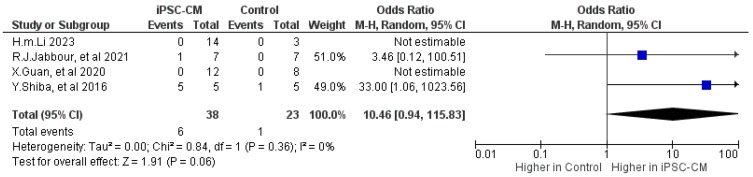
Effect size of iPSC-CM treatment on arrythmia [9,19,30,31]. Blue box represents the point estimate of the effect for a single study, the black diamond represents the overall effect estimate of the meta-analysis.

**Figure 5 ijms-25-00987-f005:**
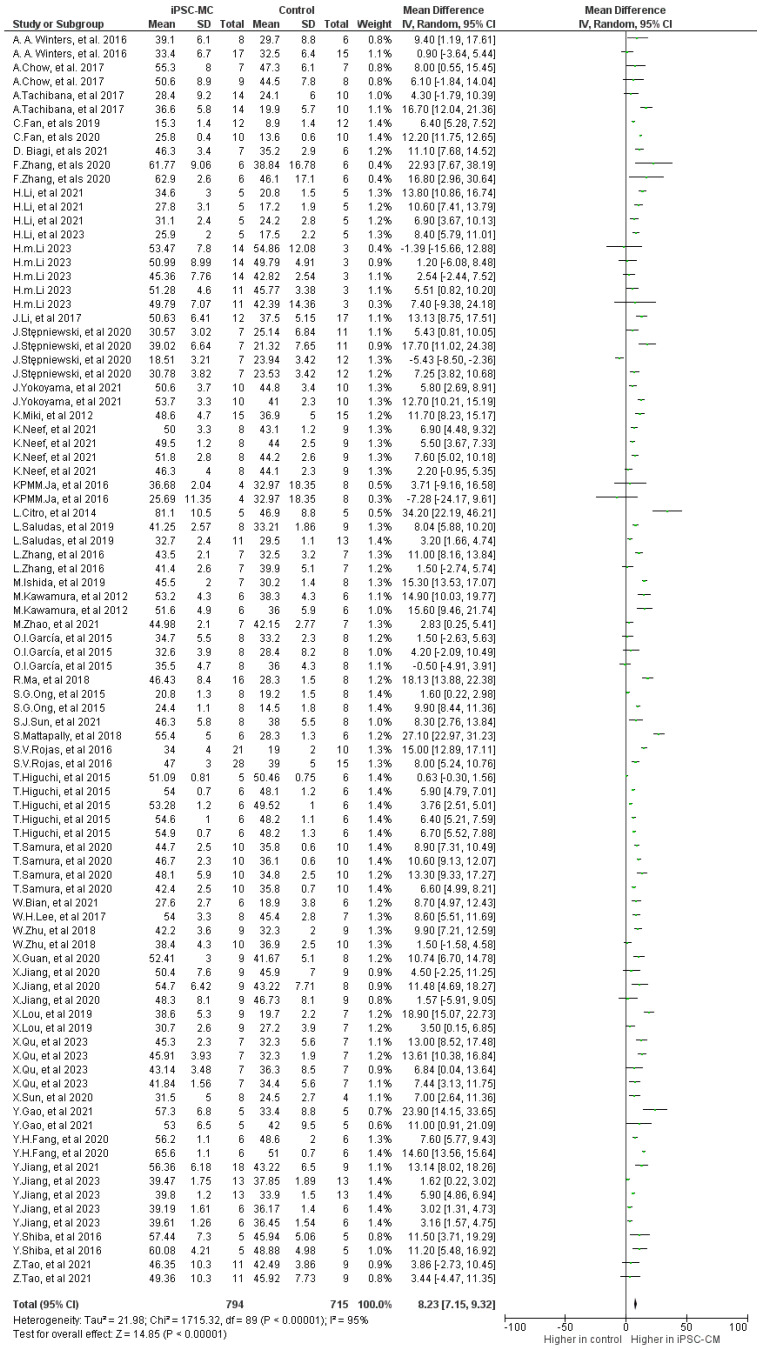
Effect size of iPSC-CM treatment on ejection fraction [9,10,12,13,14,16,17,18,19,20,21,22,23,24,25,26,28,29,30,32,33,34,35,36,37,38,39,40,41,42,43,44,45,46,47,48,49,50,51,52,53,54].

**Figure 6 ijms-25-00987-f006:**
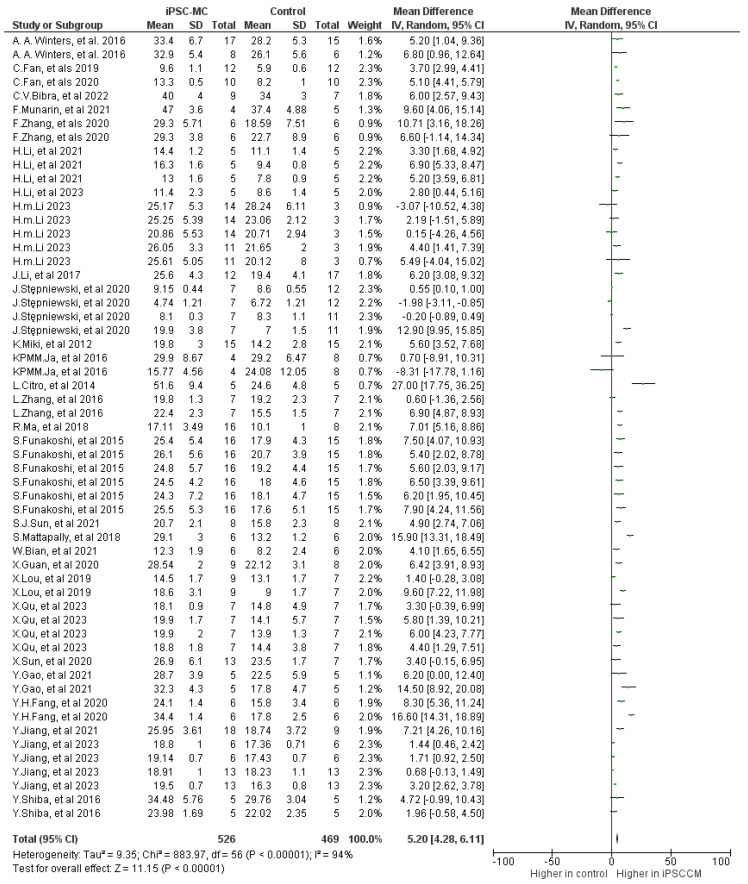
Effect size of iPSC-CM treatment on fractional shortening [9,13,14,15,18,19,24,26,27,28,29,30,32,33,36,37,38,39,42,44,46,48,50,51,52,53,55,56,57].

**Figure 7 ijms-25-00987-f007:**
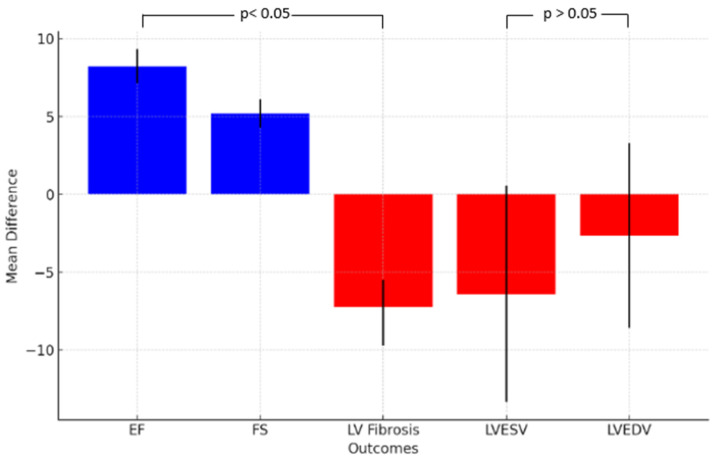
Summary of meta-analysis’ outcomes. The blue columns represent enhanced effectiveness, the red columns represent reduction effectiveness.

**Table 1 ijms-25-00987-t001:** Effect size estimation for LVESV, LVEDV, and LV fibrosis comparing iPSC-CM therapy vs. placebo.

Outcome	Number of Included Studies	Number of Treatments/Controls	Mean Difference (95% CI)	*p*	I^2^
LVESV (µL)	13	114/139	−6.41 (−13.36 to 0.54)	0.07	96%
LVEDV (µL)	16	199/191	−2.66 (−8.6 to 3.27)	0.3	94%
LV fibrosis (%)	30	332/276	−7.62 (−9.72 to −5.52)	<0.001	94%

LV: left ventricle; LVESV: left ventricular end-systolic volume; LVEDV: left ventricular end-diastolic volume.

**Table 2 ijms-25-00987-t002:** Subgroup analysis to explore source of heterogeneity on changes in LVEF.

Subgroup	Number of Included Studies	Number of Treatments/Controls	Mean Difference (95% CI)	*p*	Subgroup I^2^	Between Group I^2^
Animal size						
Small	36	676/663	8.33 (7.18 to 9.47)	<0.001	95%	0
Big	5	100/52	8.38 (4.03 to 12.72)	<0.001	0	
Time of follow-up (weeks)						
<4	24	333/309	5.38 (4.13 to 6.63)	<0.001	91%	94.7%
4–8	36	339/318	11.23 (9.48 to 12.61)	<0.001	91%	
>8	10	104/88	7.26 (4.39 to 10.13)	<0.001	93%	
Delivery method						
Intramyocardial	31	549/510	8.09 (6.79 to 9.39)	<0.001	92%	90.8%
Intravenous	1	25/15	2.34 (−0.74 to 5.43)	0.14	1%	
Intracoronary	1	9/9	−5.05 (−10.74 to 0.65	0.08	0%	
Bio-engineered tissue	14	262/265	9 (7.2 to 10.80)	<0.001	97%	
Treatment timing (week)						
<1	32	603/528	8.21 (6.93 to 9.49)	<0.001	93%	96.8%
1–4	10	172/167	7.15 (5.66 to 8.63)	<0.001	92%	
>4	2	19/20	15.28 (13.67 to 16.88)	<0.001	0	
Disease model						
Permanent injury	39	701/671	8.37 (7.24 to 9.50)	<0.001	95%	0
I/R	4	93/44	6.78 (2.65 to 10.91)	0.001	93%	
Cell origin						
Xenogeneic	37	635/575	8.75 (7.53 to 9.98)	<0.001	94%	80.2%
Allogenic	6	159/140	6.23 (4.40 to 8.06)	<0.001	93%	

I/R: ischemic-reperfusion.

**Table 3 ijms-25-00987-t003:** Meta-regression of potential modifiers of EF.

Variable	Coefficient	Standard Error	t-Value	*p*-Value	95% Confidence Interval
Animal size: big	−1.649	2.342	−0.704	0.483	−6.304 to 3.007
Animal size: small	3.513	2.916	1.205	0.232	−2.283 to 9.308
Treatment timing: acute	−3.108	2.266	−1.372	0.174	−7.611 to 1.396
Treatment timing: chronic	8.166	4.652	1.755	0.083	−1.081 to 17.413
Treatment timing: sub-acute	−3.195	2.211	−1.445	0.152	−7.590 to 1.201
Method delivery: Bio-engineered tissue	3.917	2.143	1.828	0.071	−0.343 to 8.178
Method delivery: Intracoronary	−8.065	3.327	−2.424	0.017	−14.678 to −1.452
Method delivery: Intramyocardial	3.824	1.641	2.331	0.022	0.563 to 7.085
Method delivery: Intravenous	2.188	2.801	0.781	0.438	−3.379 to 7.754
Cell origin: Allogenous	−0.150	1.139	−0.132	0.895	−2.413 to 2.113
Cell origin: Xenogeneic	2.014	0.922	2.185	0.032	0.182 to 3.846
Disease model: IR	1.962	2.619	0.749	0.456	−3.244 to 7.168
Disease model: Permanent injury	−0.098	2.155	−0.046	0.964	−4.382 to 4.186
Follow up: <4 weeks	1.011	1.299	0.778	0.438	−1.570 to 3.592
Follow up: 4–8 weeks	1.052	1.269	0.829	0.410	−1.471 to 3.574
Follow up: >8 weeks	−0.965	2.454	−0.393	0.695	−5.843 to 3.913

**Table 4 ijms-25-00987-t004:** Ongoing clinical trials about iPSC-CM in patients with IHD.

No	Registry No	Contact Author	Country	Study Phase	Intervention Model	No Patients	Starting Time	Expected Completion Time
1	NCT03763136	Jiaxian Wang	China	Phase 1 and Phase 2	Randomized	20	October 2021	May 2024
2	NCT04396899	Wolfram-Hubertus Zimmermann	Germany	Phase 1 and Phase 2	Single Group Assignment	53	February 2020	October 2024
3	NCT04696328	Takuji Kawamura	Japan	Phase 1	Single Group Assignment	10	December 2019	May 2023
4	NCT04945018	Heartseed Inc.	Japan	Phase 1 and Phase 2	Non-Randomized	10	September 2023	March 2024
5	NCT05566600	Jiaxian Wang	China	Early Phase 1	Randomized	32	October 2022	July 2025

## Data Availability

The datasets are available from the corresponding author upon reasonable request.

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
