# Peer review of "Induced Pluripotent Stem Cell-Derived Cardiomyocytes Therapy for Ischemic Heart Disease in Animal Model: A Meta-Analysis"

_ijms, 2024, doi:10.3390/ijms25020987_

Round 1
Reviewer 1 Report
Comments and Suggestions for Authors
The manuscript by Quan Duy Vo et al. is focused to evaluate the potential therapeutic effect of induced pluripotent derived - cardiomyocyte (iPSC-CM) for IHD patients. The study uses meta-analysis to assess iPSC-CM outcomes in terms of efficacy and safety in IHD animal model studies, investigating iPSC therapy effects on both cardiac function and safety outcomes
About 51 studies have been considered evidencing no significant differences in mortality and arrhythmia risk between iPSC-CM treatment and control groups.
Thus iPSC-CM therapy seems to be safe and beneficial for enhancing heart function in IHD, even if an evident heterogeneity among studies is present.
The first thing that I cannot understand is why the title (Induced Pluripotent Stem Cell-derived Cardiomyocytes Therapy for Ischemic Heart Disease in Animal Model: Systematic Review and Meta-analysis) reports both “systematic review and meta-analysis” words considering that a meta-analysis includes a systematic review, but a systematic review does not imply a meta-analysis.
Moreover for the meta analysis the studies analyzed must be sufficiently free from bias, uniform and not too dissimilar in design and outcomes (homogeneous), so that they can be compared. It doesn’t seem that the 51 studies evaluated for this work can be considered homogeneous, as also reported by authors
I think that for all the parameters reported (mortality, Arrhythmia, Ejection fraction and Fractional shortening) the best approach would be to not consider all 51 studies but rather to select the works as best as possible, creating a more homogeneous condition. This approach, despite a reduction in case studies, would allow for a better statistical value. This methodology will avoid the risk of heterogeneity and substantial inconsistency. Thus I think that this study should be reformulated for achieving a better statistical significance.
Comments on the Quality of English Languagethe english just needs minor revisions
Reviewer 2 Report
Comments and Suggestions for Authors
Vo et al.'s article presents a comprehensive review and meta-analysis exploring the potential of induced pluripotent stem cell-derived cardiomyocyte (iPSC-CM) therapy in treating ischemic heart disease (IHD) within animal models. Overall, the article maintains a well-structured narrative. I only have a few comments (in no order of magnitude).
· Consider creating an abstract figure summarizing the study's outcomes. This visual aid can provide readers with a quick and clear overview of the findings.
· Incorporate a table before the summary section to outline ongoing clinical trials (e.g., NCT04396899). This addition can offer valuable insight into the current state of clinical research in this field.
· In the discussion section, address the limitations associated with extrapolating findings from small animal models to those obtained from larger animal model studies. Discussing the differences between these models can add nuance to the interpretation of the results.
Round 2
Reviewer 1 Report
Comments and Suggestions for Authors
the authors' replies were satisfactory